# Development of an Immersive VR Experience Using Integrated Survey Technologies and Hybrid Scenarios

Francesco Gabellone

National Research Council, Nanotec, Via Monteroni, 73100 Lecce, Italy; francesco.gabellone@cnr.it

**Abstract:** The paper was aimed to promoting and improving the knowledge of the Naples city's monuments through an immersive visit experience, according to the paradigms of new digital languages. Thanks to the use of integrated technologies, some monuments of the city are presented in virtual way, with unusual viewpoints, that reveal previously unseen details, many of them not directly visible to tourists. A journey created by the use of integrated technologies, to discover historical facades and museums to be explored in total freedom, without physical constraints, without cognitive barriers. The technological basis supporting the visit consists of integrated solutions including digital photogrammetry, 3D modelling, virtual restoration and persuasive storytelling, all organised to provide a product for the general public, to be enjoyed with VR headsets. The available contents are organised on different reading levels, in according to three paths that include: a visit to the MANN (National Archaeological Museum of Naples), a visit within the virtual room dedicated to the most important museums of the city and a virtual walk through the decumani, the heart of historical centre. The virtual enjoyment of contexts no longer visible in original state or not accessible by tourists is resolved by the virtual reconstruction and re-location of artefacts in a virtual space, here called Virtual Room.

**Keywords:** virtual reconstruction; immersive archaeology; immersive VR

## 1. Introduction

### 1.1. Integrated Survey Technologies for VR Immersive Experience

The exponential development of VR technologies for immersive experience has imposed a consequent adaptation of rendering and representation methods and techniques. While the massive resources available in the VR development environments make it possible to create hyper-realistic "ideal" scenarios with less effort than in the past, we must note the equally growing need to obtain replicas of "real" contexts with the same ease and the same level of complexity. Such an achievement was until recently quite challenging, because survey techniques for real spaces were confined to the restitution possibilities obtainable with laser scanners and, to a limited extent, with older photogrammetric techniques. It is precisely the vertical development of the latter technologies that has opened a new chapter in the complexity and level of achievable realism, effectively establishing a new and higher level of involvement of the VR experience, given by the realism of textures, the level of detail and verisimilitude of settings. It must be noted, however, that the contribution of a single technology rarely guarantees a comprehensive solution to the many problems involved in restoring a site [1]. Laser scanning often produces excellent point clouds, but these require additional steps for necessary processing with surfaces suitable for VR use. Point cloud processing generates meshes rich in detail, metrically accurate, but often with textural quality levels not comparable to what can be obtained with modern photogrammetric techniques. In addition, the exclusive use of laser scanning often poses operationally unsolvable problems, related to the positioning of gripping stations, such as surveying from above, or from moving survey points, and so on. These limitations of use are all easily solved by adopting modern close-range photogrammetric techniques, using

normal cameras or with a drone (RPAS—Remotely Piloted Aircraft System). In most cases these techniques are faster, easier to use and, above all, cheaper than using a laser scanner. This explains the enormous development of such applications in surveying at all scales of representation, from topographic surveying to the detailed surveying of small objects. Nevertheless, a series of problems can be listed that are also present in the latter category of surveying. For example, the survey of gratings, thin elements, highly reflective objects, transparent objects, etc.

The solution to these is provided by other surveying techniques, first and foremost manual remodelling, solid modelling, surface modelling, polygonal modelling and all the remeshing operations that allow for the reworking of meshes obtained by photogrammetry or laser scanning, in order to adapt them for porting to VR platforms. Without considering the various possible integrations between photogrammetry and laser scanning, now almost always necessary in complex surveys.

From all these considerations, it is clear that a VR scenario that meets professional-level requirements for verisimilitude, realism, level of detail and portability will necessarily have to be populated by 3D models that come from diversified techniques. All of these elements must be properly integrated in order to make up for the various problems just described, without compromises that reduce the perceived quality of VR scenes. To this end, various examples of integration of survey techniques will be described in this article, with pointed references on the different issues emerged during the survey campaigns, documented here with images and links to available resources on the web related to this project. Many minor or major modifications to the produced 3D models were suggested by the perception obtainable during the VR visit, of course highly immersive, for the adoption of 3D models that can be visited in real time (Figures 1 and 2).

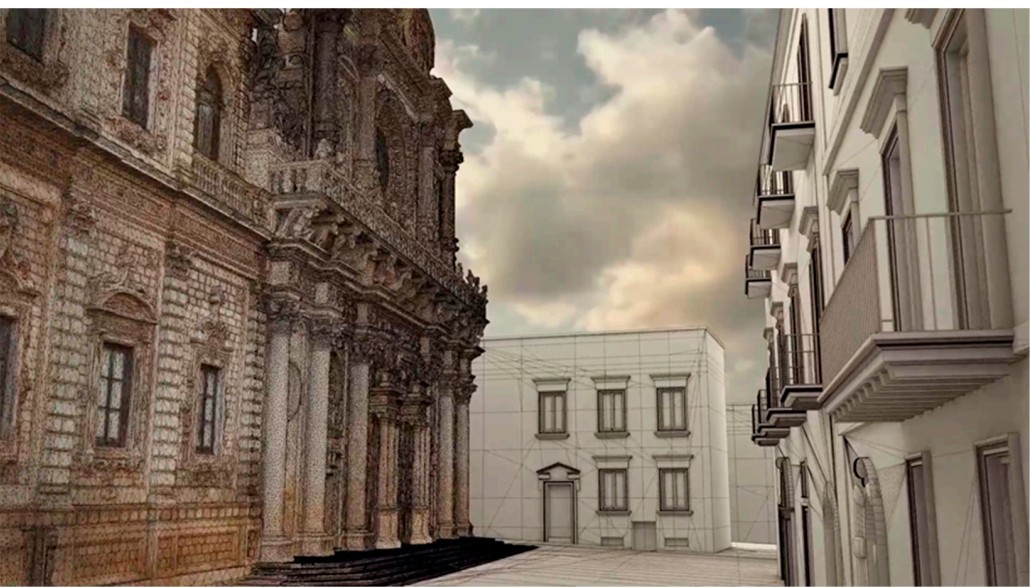

**Figure 1.** Lecce, the Basilica of Santa Croce. Example of integrated 3D survey.

*1.2. Hybridization of Media for VR Exploration in Cultural Heritage Field*

The rules of communication have long entered the digital environment, but they are social rules that existed even before the digital. Therefore, it is not the digital that determines them. In this sense, understanding digital means understanding human behaviour. Chadwick, in 2013, wrote a book entitled "Hybrid media system. Politics and power" [2], in which he argues that the media system is constantly evolving, and this does not constitute a revolution, because nothing is ever generated from scratch. On the contrary, there is a continuity between new and old media. Chadwick argues that a holistic approach must be taken toward media, that is, considering them as an interconnected and interdependent whole, since media intersect and combine with each other, producing in turn new rules. By

"logic" we mean technologies, genres, norms, behaviours and organizational forms. Each medium has its own logic, which follows rules, which must be understood and followed. When the logics of the different media come into contact with each other, hybridization is produced. With the term "hybridization", Chadwick therefore means the product of integration between old and new media logics in digital environments, i.e., communication processes that integrate logics, channels and media, in turn producing different logics, channels and media [3].

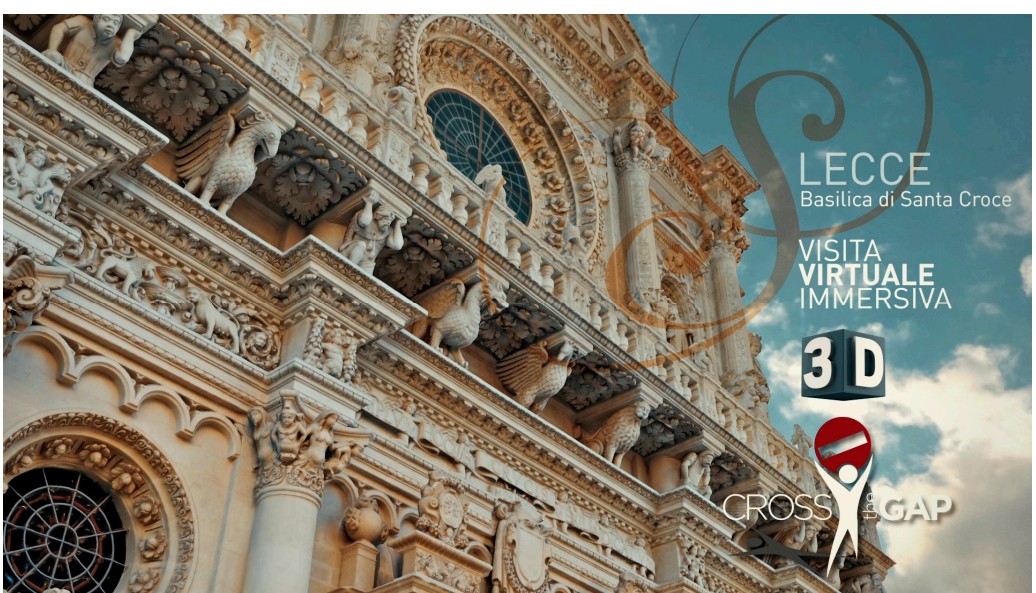

**Figure 2.** Lecce, the Basilica of Santa Croce. Example of integrated 3D survey. Screenshot of video trailer.

The hybrid media model rejects dichotomies, moves us away from the "either this, or that" models of thinking, and closer to "not only, but also" ones, drawing attention to the communicative flow. What the author is saying, then, is to look for the spaces of intersection and overlap between one medium and another, where different formats and content intersect. In this sense, then, the problem of the communication professional is how to contaminate them with each other. Approaching the world of interaction, and thus of multimedia systems for knowledge and virtual visit, Randall Collins [4] argues that the main goal of the interactant is "the successful outcome of the ritual", so a solution aimed at obtaining a positive emotional reward should be sought. Generating and increasing emotional energy, constitutes a fundamental capital of interaction in daily life, but also in virtual interaction, in the use of games and digital tools for learning. Collins speaks of strength, confidence and enthusiasm, this emotional energy is the resource that activates interaction, which in turn creates and reinforces group identity through rituals, repeating forms of interaction patterns that are effective in generating interest. An remarkable repercussion of this aspect can be found in a form of hybridization of the Virtual Museums, where the narrative construction is manifested through a marked emphasis on the use of meta-stories, narratives that highlight key points of discourse through a framing mechanism. According to this approach, narrative devices such as metaphor do not represent a mere linguistic fact, but have a strong cognitive value, constituting a way of understanding one aspect of a (usually abstract) concept in the terms of another (usually concrete) concept. This facilitates understanding and emphasizes some aspects of the first concept, while hiding others. For this reason, the metaphor also has a strong value in structuring the cognition of concepts, i.e., the way people perceive them. Metaphors [5,6] and narratives have therefore become a crucial aspect in the construction of VR-based models of knowledge, in which the hybridization of the media brings together different representative forms in order to engage and guide the visitor through a cultural experience, different from the real visit. Sense

of presence, emotional involvement, effective storytelling and multisensory solicitations are essential elements in the creative process, all determining factors in fostering contact between human sensitivity and the digital, computer-created product. The reasons for such contamination are thus related to a progressive awareness of the importance of narration in cultural communication [7]. What is proposed to be achieved by the virtual processing of Cultural Heritage is a multiplication of its communicative potential, in an attempt to better transmit and make understood the cultural message that the asset carries. Virtualization, through the use of digital technologies, facilitates those mental processes of abstraction, allowing one to see beyond the visible, to make legible and recognizable what is not always easy to identify, understand and contextualize [8,9]. The hybridization of media within a VR platform actually offers new opportunities for curators and the public, through the conception and experimentation of hybrid collections, integrating physical artifacts with digital ones, using virtual and augmented reality together with storytelling strategies. The development of the platform presented here opens up interesting opportunities for inclusion and accessibility, bringing together hybrid scenarios for learning, communication and knowledge of Cultural Heritage.

## 2. Project Presentation

Financed by the Mibact's Cultura Crea project, on the initiative of the Fondazione di Comunità del Centro Storico di Napoli, this project, called Virtual Neapolis, was created with the aim to promoting and enhancing the city's monuments through an immersive VR visit experience, according to the paradigms of new digital languages. A walkthrough discovering monuments, streets and Naples museums. In this project some of the city's monuments are presented in virtual way, with unusual viewpoints that reveal previously unseen details, many of them not directly visible by tourists. This objective has been achieved using integrated technologies, managed to discover historical facades and museums to be explored in total freedom, without physical constraints, without cognitive barriers. The technological basis supporting the visit consists of integrated solutions including digital photogrammetry, 3D modelling, virtual restoration and persuasive storytelling, organized to offer an experience to be enjoyed with VR headsets. The available contents are articulated on different reading levels and are organized according to three paths (Figure 3) that include: a visit to the MANN (National Archaeological Museum of Naples), a visit to the most important museums of the city and, finally, a virtual walk through within the decumani, the heart of the historic centre. For the first two tours the virtualization based on the concept of "museum collections" has been used. Here the virtual objects have been arranged according to museographic criteria, which refer to their origin, morphology and type. They are therefore grouped together in an environment that is strongly different to the real museum space, (in which the originals are located). The original design of this virtual space is aimed to activate a visit that is very different from the one obtainable in the real museum. This exhibition space is not conceived to obtain a copy or a surrogate of the real space, but rather to offer the visitor an alternative visiting experience, in which the objects can be freely explored from any point of view, with results that can only be activated in a virtual environment.

This refers to a virtual room, a digital space modelled to organize objects and information in very readable way, conceived to promote experiential paths. The virtual room fully exploit the potential of movement and visual perception typical of a 3D environment. The programming languages, the user experiences and the navigation paths may at first glance recall the structure of a game, but the virtual room does not give scores and does not require the interaction skills that characterize games. On the contrary, it should facilitate the enjoyment and the visit, to mainly catch the attention and the emotional involvement of the visitor [10]. All determining factors in the mnemonic processes and in the understanding of the transmitted cultural messages. The virtual room is therefore an ideal room, created entirely in a digital environment, suitable for conveying cultural messages through the typical functions of an immersive and interactive environment based on virtual reality.

As mentioned above, in this project there are two virtual rooms conceived in this way, the first one is dedicated to the House of the Faun and a second one dedicated to the masterpieces of Naples museums, both described further on. The Figure 4 shows some frames of the platform, in particular the aspect of the user experience in the virtual rooms with the interaction points and the texts activated by proximity to the object. Further on are some grab-screens of the city model with the POI (Figure 5), the spheres that allow 360° stereoscopic navigation in real space (Figure 6) and the info available in the visit to the House of the Tragic Poet (Figure 7, Supplementary Materials). The three paths were developed with innovative VR-based solutions.

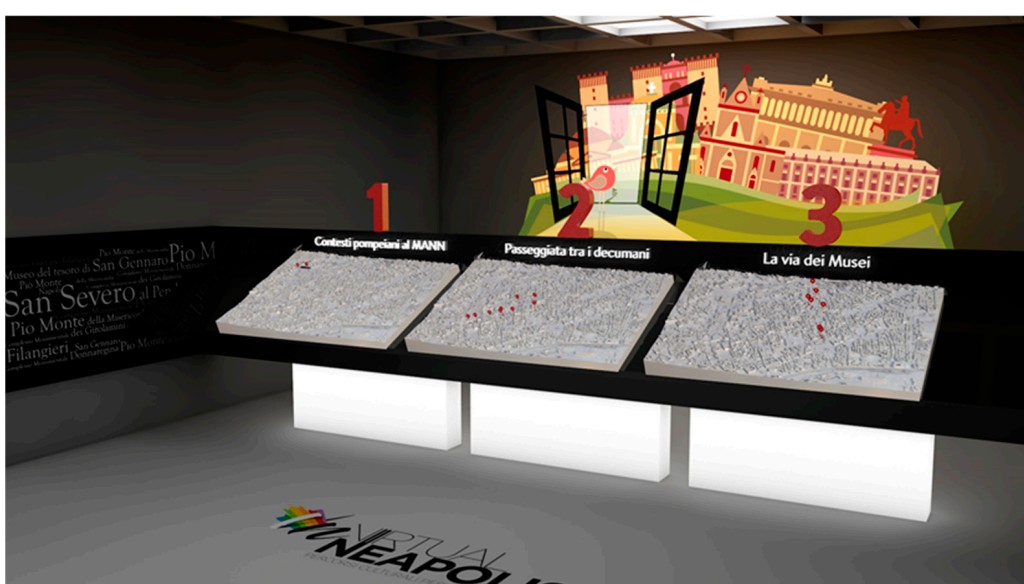

**Figure 3.** The entrance of the VR platform with three paths.

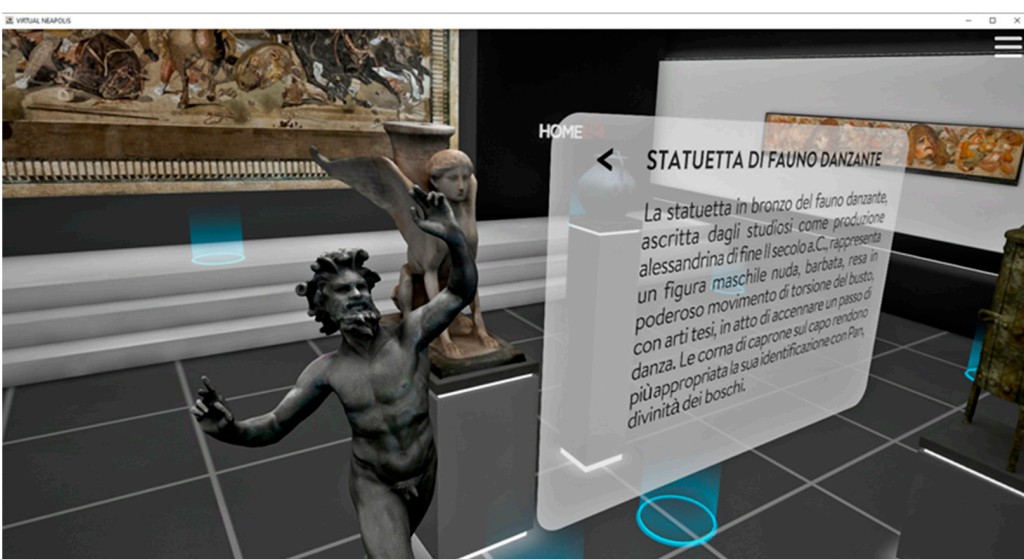

**Figure 4.** The Virtual Room of the House of Faun with the interaction pints and text activated with proximity trigger.

As is well known, VR technologies are currently at the centre of interest of a large scientific community, especially in relation to the possibilities related to the distant visiting of cultural heritage and the delivery of scientific and educational content. Some studies have evaluated how effective the use of virtual reality is in improving attention in cognitive training programs [11]. VR technology been shown to be effective in knowledge transfer

processes but encouraging results have also been reported in the behavioural disorders and social relationship problems of the analysed subjects. Immersive virtual reality used in cognitive training has been shown positive results in improving attention. It can improve the ability to concentrate of children and adolescents with behavioural problems and thus help them in learning. These results are interesting and perhaps partly unexpected, because the development of VR enjoyment started mainly with purposes related to the game and simulation industry. With the advent of new 3D engines, the overall performances have gradually ensured the porting of more and more realistic environments, which allow a greater interaction with users and above all more realism in the immersive experience [12]. Starting from these premises, the Fondazione di Comunità del Centro Storico di Napoli launched a project, funded by Invitalia, based on the potential inherent in VR systems, in which cultural paths can promise different heterogeneous realities in order to present the extraordinary treasure of the city's cultural heritage, but in an innovative way. The platform should also constitute an element of territorial marketing and provide employment opportunities for young operators in the cultural tourism sector.

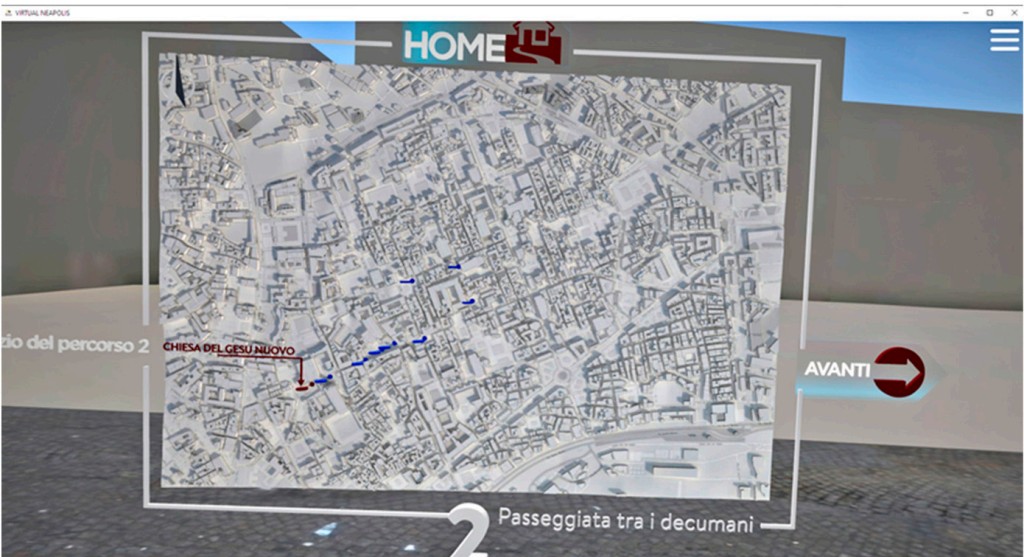

**Figure 5.** The city model with the POI and navigation button.

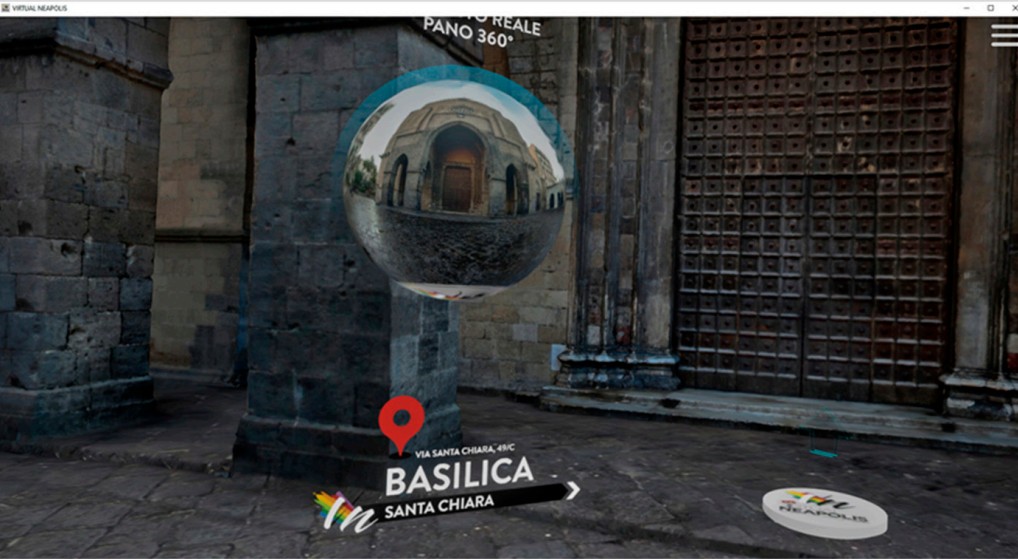

**Figure 6.** The info related to the visiting point and the sphere with the possibility to visit the real space, with stereoscopic panes.

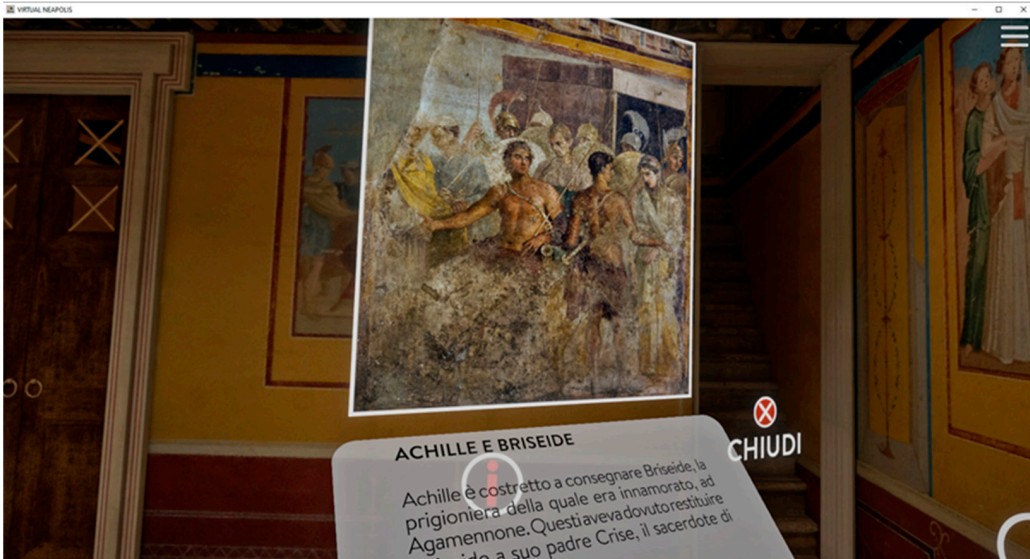

**Figure 7.** The informative sheet during the visit to the House of the Tragic Poet.

## 3. Methods and Technologies

*Survey Technologies, Current Tendencies*

The technology behind this virtual experience is fundamentally centred on the massive use of digital photogrammetry. This is a well-known 3D acquisition method, often widely used for its low cost, for speed and the possibility of generating photorealistic models complete with high-definition textures. Photogrammetry or image-based restitution, as the name suggests, derives measurements and 3D models from photographs. In recent years have seen the vertical development of this technique and the subsequent development of new restitution methods in the field of terrestrial photogrammetry. Low-cost commercial software packages, based on automatic or semi-manual measurement, make it possible, after an orientation and bundle adjustment phase, to obtain a set of calibrated data from a series of images [13]. The evolution of this software has been very fast and promising in recent years, to the point that today many studies and research laboratories prefer these to other active techniques, such as laser scanning. Only a few years ago, it was difficult to venture into the acquisition of 3D models from photos. Most operators in the sector preferred to use a laser scanner, which was much more expensive, but very accurate. Photogrammetry, on which all these image-based techniques depend, required a lot of effort to contain measurement errors. The most commonly used techniques were photo-rectification, 3D generation from ordered silhouette sequences and point-by-point photomodelling. Photo-rectification was mainly used in architecture, in order to obtain orthophotoplans of facades. The procedures most commonly used today are far removed from the early experiments just described, are fully automated and start from a sequence of uncalibrated images [14]. Commercial monoscopic multi-image software uses techniques that exploit the principle of correlation and allow the internal and external orientation of the frames to be carried out automatically, then generating the three-dimensional model complete with textures. The results obtained in various experiments have shown only a few problems due mainly to the presence of "holes" and "lacunas", essentially due to the information deficit, i.e., the poor photographic coverage in those points. Other difficulties were encountered in the case of objects with too regular, poorly characterized surfaces, e.g., railings, thin or square elements. In spite of this, the new software uses flexible algorithms, all derived from the structure-from-motion (SfM) algorithm, such as to guarantee the orientation of the photos even in the absence of the classical procedures prescribed by digital photogrammetry, i.e., without prior calibration of the camera and without any substantial input from the operator in the orientation phases. All the operations are therefore automatic, leaving open the possibility of setting parameters according to

the quality desired for the definition of the number of polygons and the dimensions of the textures to be generated [15–21]. As already mentioned, poorly characterized objects are very difficult to render, just because the software is not able to "trace" any features, i.e., it is not able to "follow" well distinguishable points. On the contrary, dirty or old objects are very easy to render. This explains the success of this technique in archaeological survey campaigns and in our case of the survey of historical facades (Figure 8).

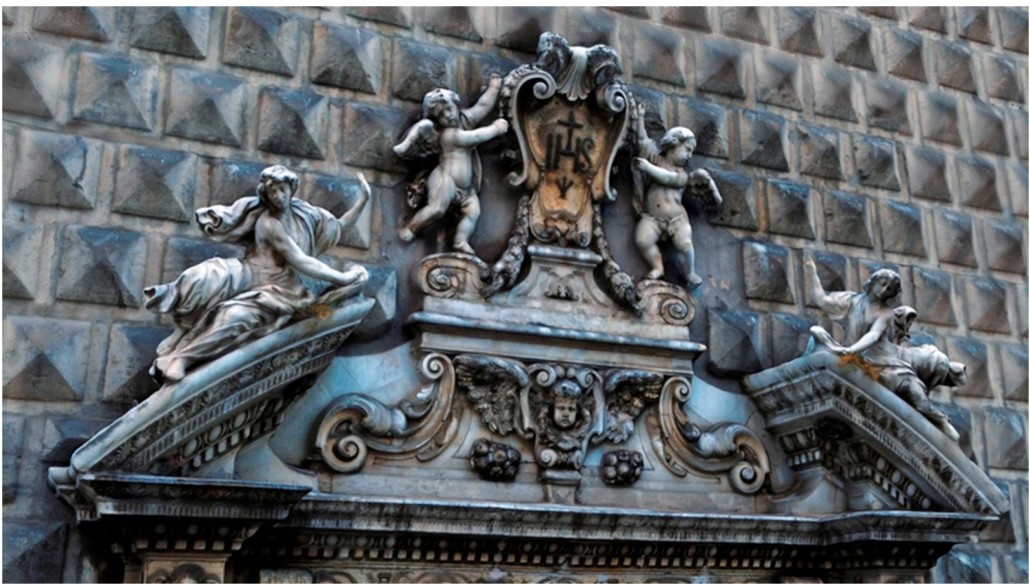

**Figure 8.** Detail of the sculptural apparatus on the church portal of the Chiesa del Gesù Nuovo.

In this specific work, the photogrammetric survey campaigns were conducted mainly with an ultralight (inoffensive) drone, for reasons of compatibility of use in confined spaces and for the possibility of flying over densely populated areas for limited times. The adoption of this method is due to the impossibility of making the surveys with other indirect survey techniques. The use of a laser scanner would have given several important problems in the acquisition of the top parts of the buildings. It would have given many problems on the undercuts, problems related to the mapping of visible colour (texturing), to the resolution and quality of the photos and, last but not least, the presence of urban environmental "disturbance elements", such as crowding, the presence of birds, cars in transit, electrical cables, etc. The ultralight drone used for this work was demonstrated to be extremely versatile. The sensor, 21 MP wide angle (5344 × 4016)/4:3/84° HFOV and 12 MP rectilinear (4000 × 3000)/4:3/75.5° HFOV, was adequate for the purposes of the project and its low resolution did not lead to significant morphological deficits. It should be remembered that the resolution of the individual photos from which a photogrammetric model is generated is not in itself an impediment to obtaining accurate 3D models. Satisfactory results are obtained by increasing the number of photos and becoming closer to the object to be rendered. A further element of merit in the use of these sensors, compared to an SLR lens, is their high depth of field, which at the operational level almost always ensures that the objects being photographed are correctly focused [20]. This aspect considerably improves the overall quality of the rendering and above all the sharpness of the generated textures. A very critical element in the survey was observed in the precise restitution of the building's grids and railings. A very common problem even when using a laser scanner for architectural surveys. The problem encountered relates to the resolution of the meshes and to the number of shots taken on these elements. In order to correctly resolve this problem, a large number of targeted shots must be carried out, from various points of view, with obvious costs in terms of time and computational costs. In our specific case, in consideration of a repeatability of the modules that characterize the railings, we carried out a precise survey only on some basic elements, which were subsequently subjected to

retopology. The software used for this operation is available free of charge from this link https://www.3dart.it/free-auto-retopology-with-instant-meshes/. The individual modules well reconstructed were then duplicated and placed in the correct position (Figure 9).

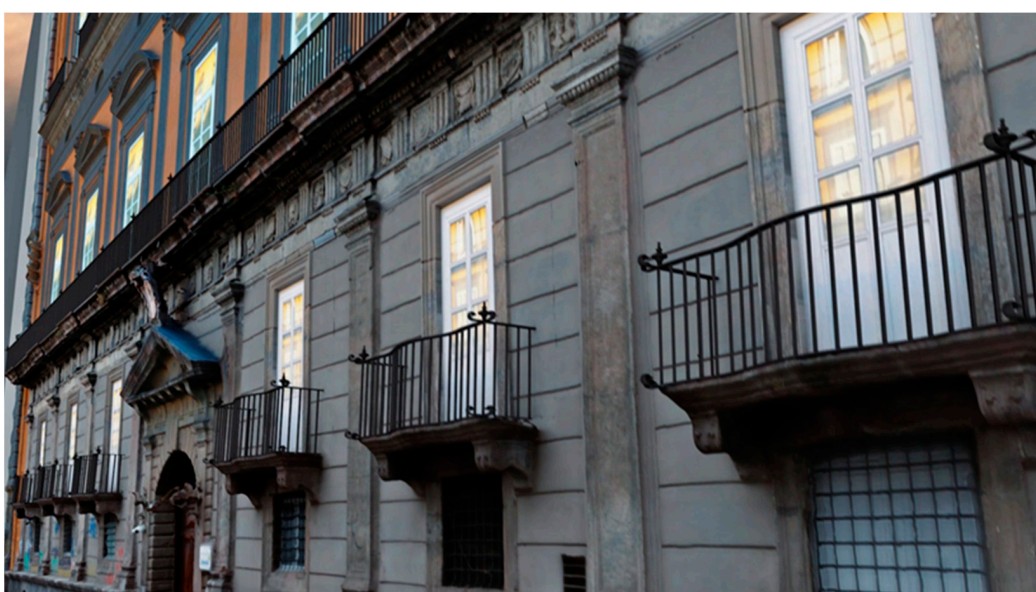

**Figure 9.** Example of restitution of the building's grids and railings with integrated techniques.

The VR platform as a whole then integrates three-dimensional models mostly created through photogrammetry, either by drone or ground camera. A particular approach was used for the 3D restitution of the model of the House of the Tragic Poet. This is a Roman house in Pompeii, dating to the 2nd century BCE. The house is famous for its elaborate mosaic floors and frescoes depicting scenes from Greek mythology. Discovered in November 1824 by the archaeologist Antonio Bonucci, the House of the Tragic Poet has interested scholars and writers for generations. Although the size of the house itself is in no way remarkable, its interior decorations are well known for the highest quality among other frescoes and mosaics from ancient Pompeii. The house is currently closed to the public. The main entrance, in which the famous mosaic with the inscription Cave Canem is visible, is closed by a glass plate. Since it was not possible to access the interior to carry out surveys in situ, we decided to adopt a different approach, perhaps more original than the "simple" survey. As already mentioned above, we have included in the virtual tour an immersive path inside the wooden model of the same house, the work of Enrico Salfi, currently kept at the MANN (National Archaeological Museum of Naples).

The virtual visit is therefore conceived as a moment of immersive enjoyment inside the model, which returns the interiors as they were visible at the beginning of the last century. The survey followed two distinct phases: in the first phase, a rough 3D model was created by the means of photogrammetry, useful to simplify the remeshing phases, in the second phase the house was remodelled manually. In fact, the restitution of the model with only the photogrammetric technique would have required a very long time, especially for the impossibility of solving very small and hidden details. For this reason, it was decided to reshape the house manually, in a 3D software (Maxon Cinema 4D). The rough model, based on photogrammetry, was then used as a referring base [22]. On this were redefined all the internal surfaces, textured with the individual photos shooting on every part of the model. The photos required for texturing were taken from the front in order to facilitate subsequent rectification operations. Since the wooden model is, except for some details, geometrically referable to flat surfaces, this operation was not particularly complex and produced satisfactory results.

Within virtual visit platform, the facade models of historical buildings have been inserted in the urban environment. This has been synthetically rendered in neutral colour.

In the virtual rooms all objects have been transformed with bake textures, i.e., with shading, radiosity calculation and environmental occlusion on pre-calculated textures. The Figure 10 shows the results of this technique within the platform, in real-time mode. The church of Gesù Nuovo with its portal decorated with sculptures (Figure 8), the Mausoleum of Mary of Hungary visible from different heights and church of the Anime del Purgatorio ad Arco (Figure 11), very complex façade, characterized by a very rich and lively baroque style. This trick allows a considerable reduction in real-time processing, also in consideration of the high number of polygons to be managed. The pre-calculated textures were subdivided into patches of dimensions compatible with the authoring software, which prefers UVW mapping with total pixel content supplied with dimensions in base 2 powers, preferably 4096x4096 pixel. The VR authoring software is Unity 3D, which has demonstrated excellent management capabilities and overall stability in relation to the target headset visor for development (HTC Vive). The use of an integrated approach between the software development and the management of a dedicated hardware allowed to maximize the performance of the system in order to guarantee the highest comfort levels achievable by the technology [17]. It was thus possible to achieve framerates of between 90 and 120 frames per second in stereoscopy, while maintaining high fidelity of detail in the 3D models. This approach was accompanied by animation and movement management designed to avoid or limit the motion-sickness effects that can occur when using these systems. The choice of a system with 6 degrees of freedom in the tracking of movements has allowed to perceive a greater naturalness and comfort during the use of contents. Finally, the technical specifications of the entire HW/SW platform make it possible to induce in the user a real and natural sense of presence in the places [16].

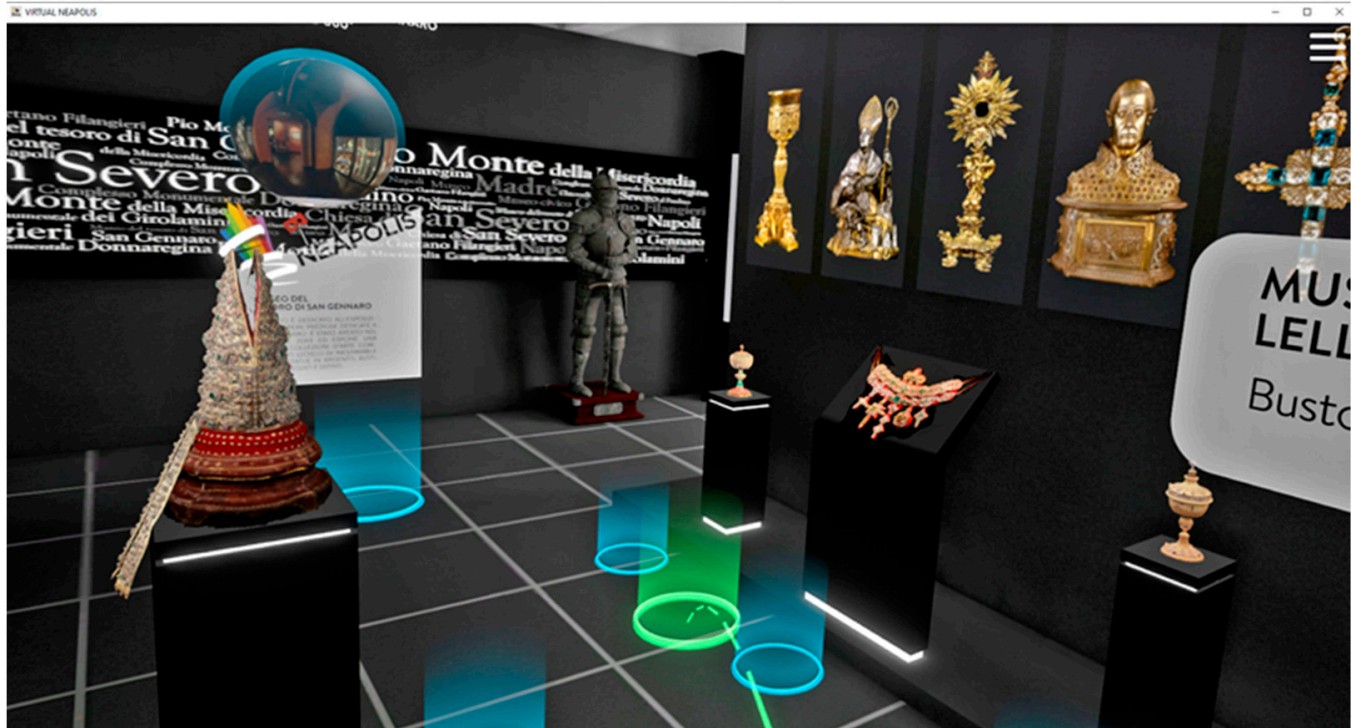

**Figure 10.** The user experience and the virtual rooms with precalculated shadows, radiosity and AO effects.

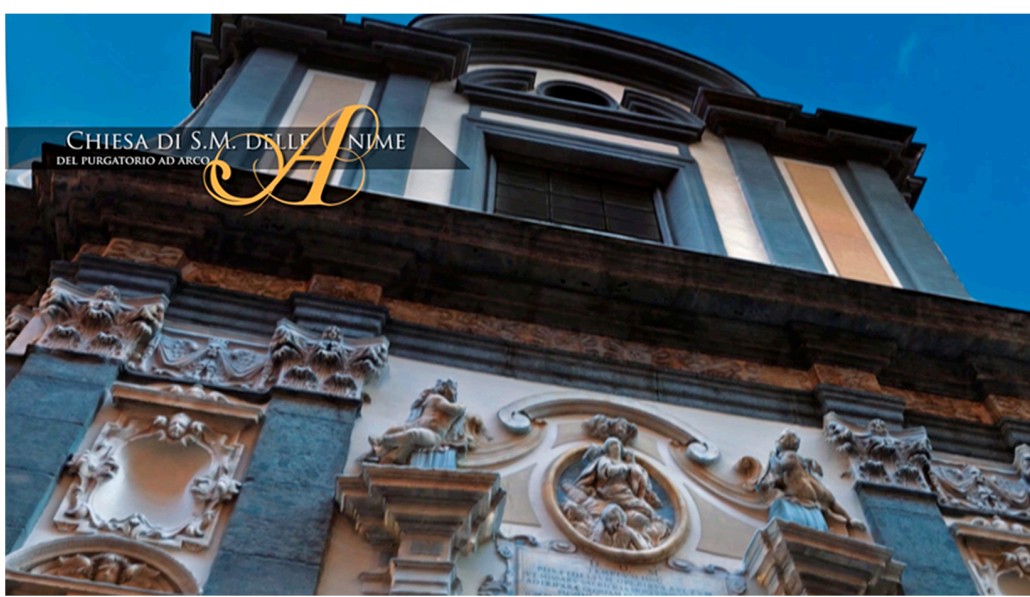

**Figure 11.** The 3D restitution of church of the Anime del Purgatorio ad Arco.

## 4. Results

### 4.1. The MANN Virtual Rooms

One of the problems often connected to museum use concerns the relationship between the object and the place of discovery, between object and context of origin. For this reason, very often the visit to museums is resolved with a purely aesthetic approach, which does not allow us to know the very reasons linked to the creation of the objects that are observed. On the other hand, the great utility of reconstructions is accepted by all, which allows for the objects to be put back in place and to appreciate not only the aesthetic values of the object, but also its function, its *raison d'etre*. For this reason, in the virtual room dedicated to the MANN, the prestigious National Archaeological Museum, objects from the House of the Faun in Pompeii are re-contextualized in their original virtual space (Figure 12). Built in the 2nd century B.C. and enlarged in the following century, the House of the Faun is one of the most beautiful, but above all large, residences in Pompeii. Its surface area covers 2970 square metres, thus occupying a large part of Regio VI, Insula 12. Entering through the tufo stone portico you arrive in the vestibule. The floor is decorated with a refined opus sectile, with triangles in marble and limestone. You then access the atrium where the centre is placed the impluvium that here is made of travertine and not tufo, as was expected.

Near this impluvium was found the bronze statue of the dancing satyr that gives its name to the house. The numerous objects from the House of the Faun, now preserved at the MANN, have been accurately digitized in 3D, thanks to which they can be explored from any point of view and enriched with information activated by proximity to the object. Their virtual view is accompanied by their relocation in the context of their origin, with a virtual reconstruction of the House in which the user rediscovers the sense of the connection between artefact and site [18].

The same approach was used for the House of the Tragic Poet, currently not accessible by tourists. For this case study, the virtualization mainly concerned the reading of the mosaics and the famous and extremely rich pictorial decoration (almost entirely preserved at the National Archaeological Museum of Naples) relating to the last period of Pompeian art. The house is so called because of the mosaic emblem (panel) inserted in the floor of the tablinum depicting the theatrical rehearsal of a satirical drama and for the presence, in the pictorial decoration, of themes and episodes taken from the Iliad. The impossibility of direct and physical access to the site suggested the adoption of an interesting expedient for the 3D restitution of this context, in some way linked to the valorisation and knowledge of the "exhibition systems" of the past. The house was excavated between 1824 and 1825,

but a few years later it was reconstructed by the painter Enrico Salfi with a plaster and wood model (Figure 13a) that reproduces its physiognomy in detail, some of which is no longer visible in situ due to conservation problems occurred after its discovery. The model is therefore a document itself, from which we can deduce detailed information about the interior decorations, the structure of the house and its planivolumetric articulation, In the VR platform the fruition has therefore shifted to a visit experience inside the wooden model (Figure 13b), in which the mosaics and frescoes displayed in the museum have been relocated. The visitor has the possibility of immersing himself in the reconstructed house according to this particular approach, with the possibility of accessing detailed information about the iconography of the figurative apparatus, viewable through the interactive activation of the high-resolution photos taken at the MANN.

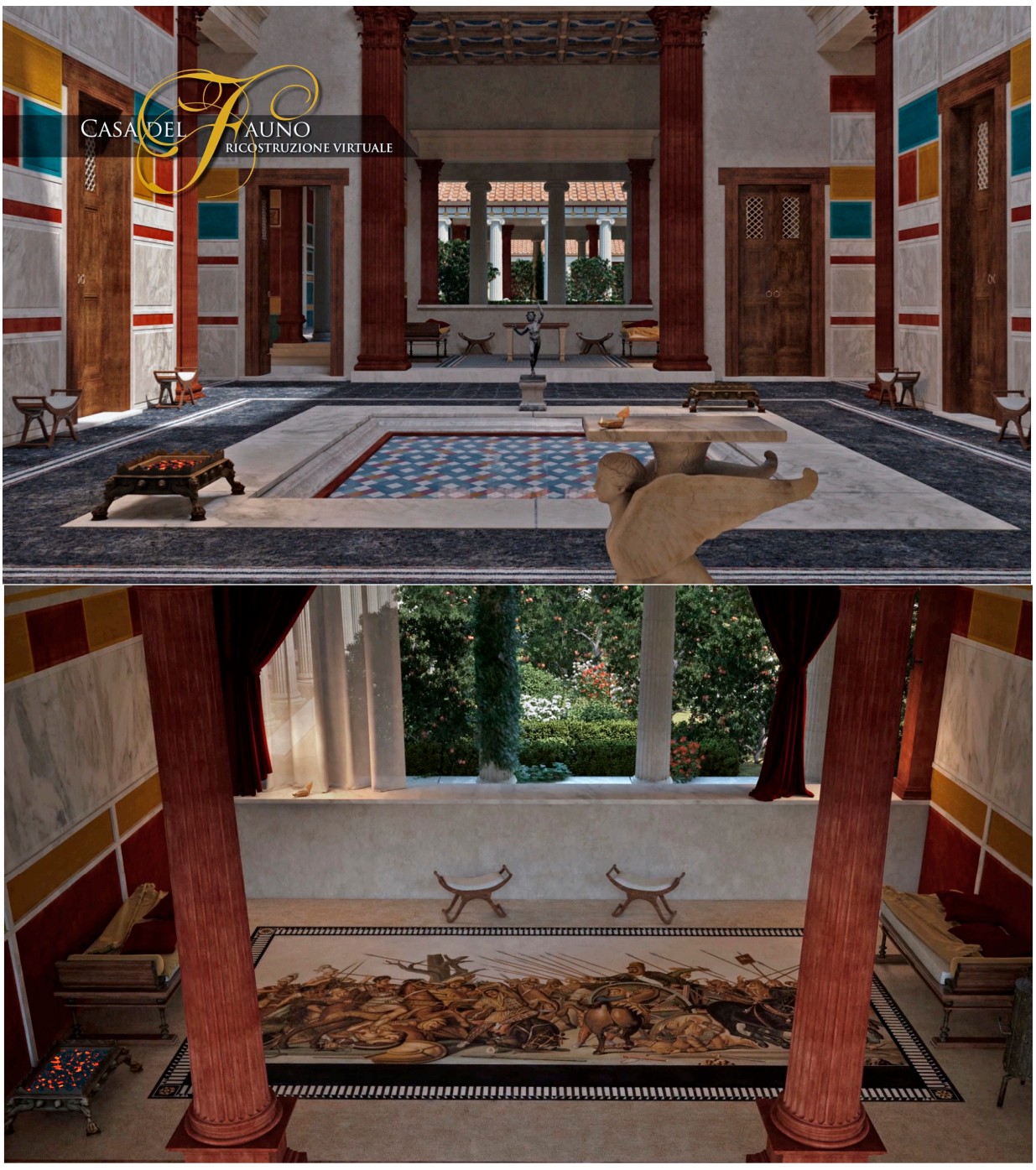

**Figure 12.** The virtual reconstruction of the House of the Faun.

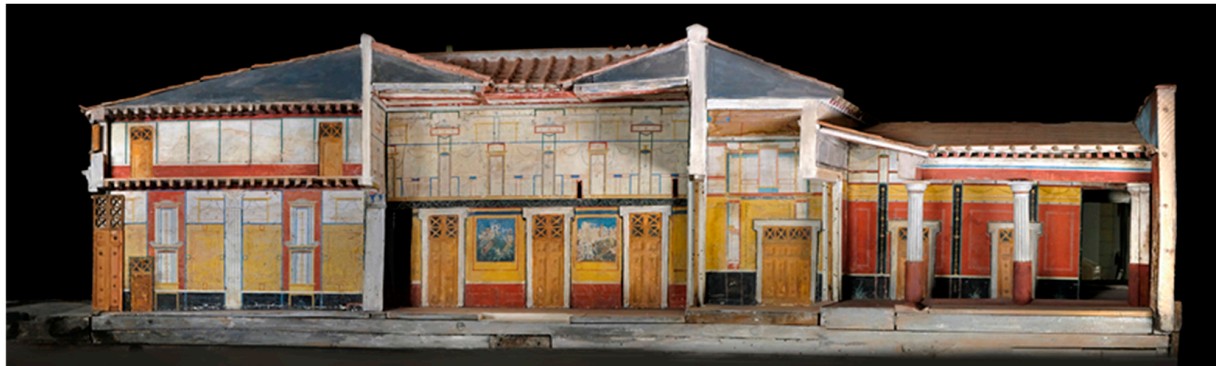

(**a**)

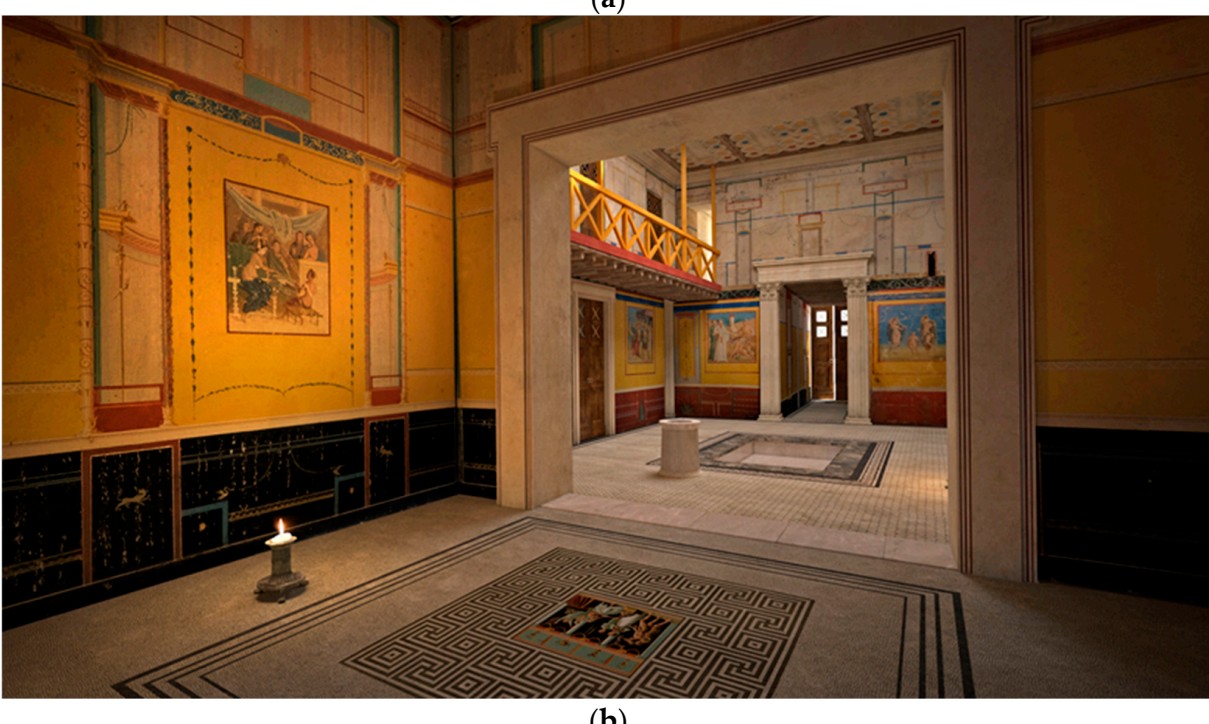

(**b**)

**Figure 13.** (**a**,**b**) The house of the Tragic Poet, the wooden model preserved at the MANN and the 3D reconstruction visible in real time mode.

*4.2. The Museums Paths Virtual Room*

The second virtual room, on the other hand, brings together significant objects from other important museums in the city: the Museo del Tesoro di San Gennaro, with its precious and unique works of inestimable value. The Museo Civico Gaetano Filangieri, with its heterogeneous collections, the Museum of San Severo al Pendino, the Diocesan Museum and the Church of Donnaregina with the mausoleum of Mary of Hungary, wife of Charles II of Anjou. In the foreground, the Cartastorie, a unique museum of the Historical Archives of the Banco di Napoli and the Seven Works of Mercy, Caravaggio's masterpiece exhibited in the Pio Monte della Misericordia complex. The virtual space is also affected here by the presence of a selection of works organised along an experiential path, where the masterpieces of the museums just mentioned are faithfully rendered through integrated technologies, with the aid of digital photogrammetry. The three-dimensional models, well-found with high-resolution colour textures, reveal to the visitor previously unseen details with unusual approaches and views impossible in reality. Especially interesting here, is the 3D transformation of a painting by Tommaso Ruiz depicting the Riviera di Chiaia in 1730 (Figure 14). The transformation of the painting makes it possible to observe the depicted scene through a three-dimensional diorama, which promotes a reading by

individual elements, an observation according to depth planes. Further along the corridor of the room, separated by the two adjacent virtual rooms, are the Scolatoi of San Severo al Pendino, which are generally closed to the public, but here can be visited without physical or psychological restrictions. Another interesting element is visible in the section devoted to the Donnaregina complex, in particular the mausoleum of Mary of Hungary. The tomb was commissioned by the King of Naples, Robert of Anjou, son of Mary of Hungary, to respect the wishes of her deceased mother to be buried in the church she herself had built. The front of the sarcophagus has seven small niches with small columns with pointed arches that contain small statues of the queen's children. In the VR platform the visitor will be able to use a virtual lift to observe the interior of the sarcophagus up close and from above, 3D reproduced in every part to not to limit the potential offered by VR technology [19].

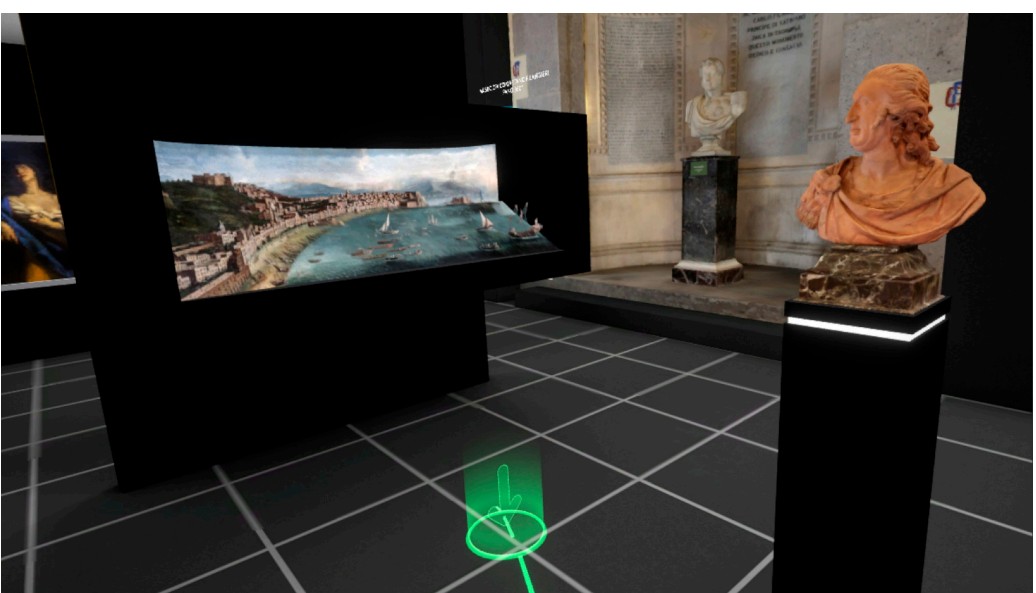

**Figure 14.** The 3D transformation of a painting by Tommaso Ruiz.

### 4.3. The Virtual Visit through City Center

The visit ends with a walk through the decumani of Naples (Figure 15). This third path has been created starting from the 3D restitution of nine historical building facades, all of which can be used without physical barriers and also viewable from unusual viewpoints. Visitors will be able to lift themselves up, to observe architectural details located at heights of 10–15 m or simply observe the monument from angles that would be impossible during a real visit. In the San Gregorio Armeno district, visitors can also observe the frieze of the church of the same name, which is embraced between the facades of the buildings fronting the street, about three metres apart. Thus, continuing the virtual tour, the visitor can admire the statue of the god Nile in the small square of the same name. The history of the sculpture dates back to Greco-Roman Naples, when numerous Egyptians from Alexandria settled in the area where the monument still stands. The Neapolitan people welcomed these colonies, later nicknamed the "Nilesi", in honour of the Egyptian river (Figure 16). The Alexandrians decided to erect a statue that would remind them of the Nile River, which had been elevated to the rank of deity bringing prosperity and wealth. Over the following centuries, however, the statue was abandoned and forgotten until it was found again in the mid-12th century. The statue, deprived of its head, was then abandoned again, only to be rediscovered in the 15th century. It is therefore an important testimony to social integration and the rediscovery of cultural values.

A final remarkable case study is the 3D restitution of the Casina Pompeiana, inside Palazzo Venezia, in the heart of Naples. Palazzo Venezia is the custodian of an ancient history. It was donated by the King of Naples Ladislaus II of Anjou Durazzo to the Serene Republic of Venice around 1412. The "Neapolitan palace of Venice", as it was called, was

the seat of the Venetian embassy in the Kingdom of Naples for about four hundred years. The building was also the object of attention by Benedetto Croce, who described it as one of the most important buildings in the heart of the city, unique in its kind. Located along the lower decumanus, just a few metres from the prestigious Church of Santa Chiara and Piazza del Gesù Nuovo, Palazzo Venezia, with its spectacular loggia and Pompeian house surrounded by a hanging garden, stands out for the intensity and strength of its history. Its verdant amenity has been the subject of numerous literary reflections, underlining its paradigmatic character and the distinctive features of the Neapolitan Garden. At the end of its recent history, in 1816, the palace was ceded by the Austrian Empire to the jurist Gaspare Capone, who created an apsidal volume between the loggia and the garden. In this way, the green area is brought back to its original condition of not immediately visible place, hidden and surprising just for the anomalous coup de theatre that it arouses in the visitor. A Latin inscription stands out on the façade of the house, almost a romantic dedication by the owner to this place, recalling the complex values it summarises and transmits:

CARA DOMUS SED UBIHORTULUS ALTER ACCESSIT QUANTO CARIOR ES DOMINO NUNC ET ADESSE AT ABESSE FORO NUNC TEMPORE EODEM VIVERE MI RURI VIVERE IN URBE LICET A. 1818.

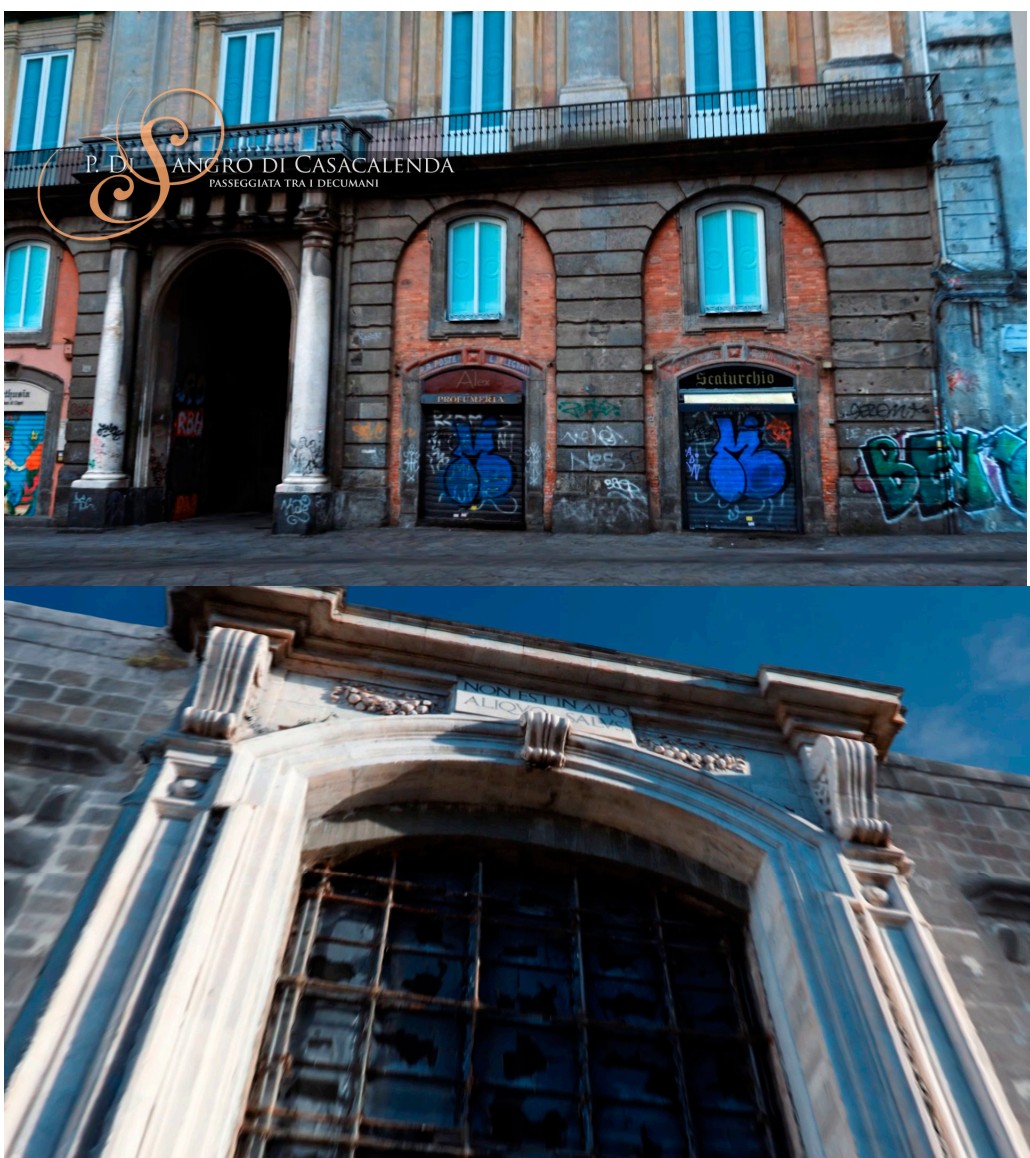

**Figure 15.** Some grab-screen of stereoscopic video trailer with the same scenes viewable within the 3D platform.

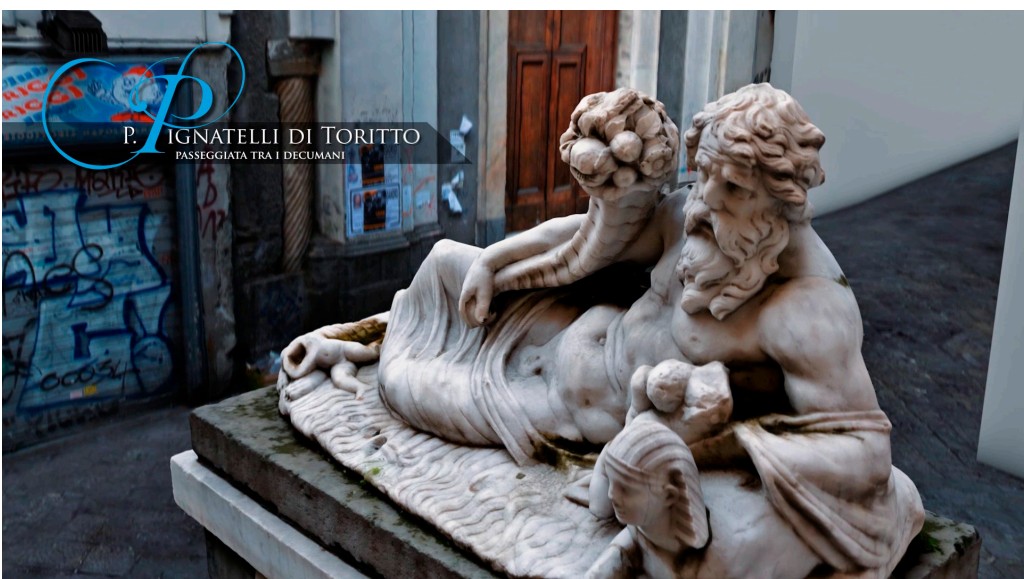

**Figure 16.** The statue of Nile, inserted in the reconstructed space, with the unusual point of view.

(For a long time you have been dear to me, O house, but since a vegetable garden has been added the dearer you are now to your master and I can now take part in public life or not take part and at the same time I can live in the country and live in the city).

The video-trailer of the Virtual Neapolis project, which passively illustrates the aims of the project, ends with the transcription of this singular dedication, written specifically to stimulate the visitor's curiosity to learn about these places in person because, as we recall once again, the virtual experience is not intended to replace the real visit, but rather to enrich it.

## 5. Discussion of Project Feasibility and Future Appliance

The main objective of this virtual visit platform, called Virtual Neapolis, is to promote the knowledge and the distant visit of the architectural, artistic and archaeological heritage of the city through a selection of monuments and contexts representative of a precise historic period, of a peculiar style. In accordance with the concept of media hybridization expressed in the opening, the term "platform" here used to describe the methodological approach, is conceived as multi content showcase, with several types of contents, such as 3D models, videos, images, reconstructions, texts, all collected in a multimodal VR use. Several type of contents for different issues, that meet different needs, but all characterized from the immersivity of vision and ease of use. The choice of Naples monuments has been evaluated in relation to their artistic appeal and their value of testimony. The virtual visit of these ancient contexts allows us to answer to different needs. First of all, it allows the vision from unusual points of view [1], allowing a reading of the figurative apparatuses of the buildings without physical barriers. This possibility makes the visit suitable also for disabled people, allowing them to overcome architectural barriers. A second very important aspect is linked to the possibility of understanding the meaning of the works displayed and their importance in relation to the period of reference. This is a way to reveal the different meanings of the works, therefore a way to increase the knowledge and interest in the artistic, historical and cultural heritage of the city. Since the increase of knowledge is the first step towards the valorisation of the cultural heritage, Virtual Neapolis promotes knowledge, but through the use of immersive technologies. A way of knowledge communication made effective by Virtual Reality, especially for the benefit of new generations [9–13]. In the Virtual Neapolis platform the used technologies are "invisible" to the users. The visit is developed in total freedom, both in seated mode and in "room scale" mode, which allows to move freely and dynamically in every direction. The visit in the Pompeian contexts introduces a new element, the user is no longer in

front of existing buildings, reproduced in 3D, but inside ancient spaces reconstructed in life, according to the methods of virtual archaeology, with scientific rigor and maximum realistic rendering. The contextualization of the finds exposed in the MANN allows to understand their exact function, their original location, the sense of scale and aesthetic values in relation to the whole, to the stylistic aspects of the domus in which they were located. In the last period the platform has been tested by different categories of users, in order to assess its usability and possible impacts on usage times. A time limit of 8 min has been set, with the possibility of increasing the duration of the visit for those who request it. The recent lockdown has unfortunately limited the use of the platform which, as is known, requires the wearing of a viewer and controller. The Community Foundation, which has the task of promoting the use of the platform, has already started a program of use of Virtual Neapolis by cultural associations that will be able to present it shortly in some spaces in the historic centre of Naples. Invitalia (the funding agency), the national agency owned by the Ministry of Economy that encourages the birth of new businesses and start-ups, has declared Virtual Neapolis a project of excellence, to be used as a pilot project for similar initiatives, to be extended to other Italian cities. We are therefore waiting for the concrete start of the presentation to the public in order to know the real impact of the project, its relapses, its possible strengths and weaknesses.

**Supplementary Materials:** Link to the House of Tragic Poet reconstruction: https://www.youtube.com/watch?v=R42ykXaQd94 (accessed on 25 January 2023).

**Funding:** This research was funded by INVITALIA, the Mibact's Cultura Crea project, on the initiative of the Fondazione di Comunità del Centro Storico di Napoli, Bando Cultura Crea cofinanziato da Invitalia SICC: IC40000010. Codice CUP: C21E18000010008.

**Data Availability Statement:** Not applicable.

**Acknowledgments:** Special thanks go to Mario Massa, General Secretary of the Fondazione di Comunità del Centro Storico di Napoli, man of great sensitivity, who was able to manage the complex dynamics of project in all its phases. No less important is the contribution of Riccardo Imperiali di Francavilla, who was also effectively and tenaciously involved in the realization of this ambitious project. The platform would not have been realized without the support of the various directors and managers of Neapolitan museums, Paolo Giulierini (MANN), Paolo Iorio (Museo del Tesoro di San Gennaro e Filangieri) also scientific supervisor of the project, Don Alfonso Russo and Nicola Ciaravola (Complesso Monumentale Donnaregina), Sergio Riolo (Museo Cartastorie), Gianpaolo Leonetti (Complesso Pio Monte della Misericordia). The VR platform was entirely programmed by Irrazionali s.r.l.s., many thanks to Matteo Greco and Maria Teresa Levante for their precious contribution and to Massimiliano Passarelli for the 3D optimizations.

**Conflicts of Interest:** The author declares no conflict of interest. The funders had no role in the design of the study; in the collection, analyses, or interpretation of data; in the writing of the manuscript; or in the decision to publish the results.

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
