# Peer review of "Development of an Immersive VR Experience Using Integrated Survey Technologies and Hybrid Scenarios"

_heritage, doi:10.3390/heritage6020065_

Round 1

Reviewer 1 Report

Manuscript is clear and well-structured, but not so relevant for the scientific community. References are datated, introdution and conclusions should be improved.
I suggest a better analysis of the state of the art on the use of 3D models for museum applications, extending bibliography which could be updated in this way. Some references are suggested below:

Fazio, L., Lo Brutto, M., Gonizzi Barsanti, S., & Malatesta, S. G. (2022). The Virtual Reconstruction of the Aesculapius and Hygeia Statues from the Sanctuary of Isis in Lilybaeum: Methods and Tools for Ancient Sculptures’ Enhancement. Applied Sciences, 12(7), 3569.
Fanini, B., Ferdani, D., Demetrescu, E., Berto, S., & d’Annibale, E. (2021). ATON: An Open-Source Framework for Creating Immersive, Collaborative and Liquid Web-Apps for Cultural Heritage. Applied Sciences, 11(22), 11062.

Barsanti, S. G., Malatesta, S. G., Lella, F., Fanini, B., Sala, F., Dodero, E., & Petacco, L. (2018). THE WINCKELMANN300 PROJECT: DISSEMINATION OF CULTURE WITH VIRTUAL REALITY AT THE CAPITOLINE MUSEUM IN ROME. International Archives of the Photogrammetry, Remote Sensing & Spatial Information Sciences, 42(2).

Moreover, can users freely access the platform? and how? this aspect is not clear.
It is advisable to deepen the description of the images (for example indicating archaeological sites and museums) in the caption paying attention to the numbering. On lines 186 and 217 figures 7 and 8 replace 1 and 2. Even punctuation on line 289 point replace comma.

Author Response

Dear colleague, I am sending you a more updated version of the article.

Regards

FG

Reviewer 2 Report

Improve the references relevant to this research.

Gaps and noise (pag. 5): include "holes" and "lacunas" also as source of "errors"

Author Response

Dear colleague, I am sending you a more updated version of the article.

FG

Reviewer 3 Report

The paper presents an interesting case study of a cultural heritage project using innovative technologies for user engagement. With three paths, the user can access an immersive visit among the monuments and cultural institutions of Naples. The paper would be very useful for international scholars and to be published on this journal. However, there is the necessity to fully revise the article because it seems to be a translation of an already published paper "Virtual Neapolis. Un’esperienza di visita immersiva in VR per le vie di Napoli" (Gabellone and Chiffi, Archaopress 2022).
Because the article is valuable, I would strongly suggest an extensive revision to modify the text, making it more fluent and not exact a copy of something else. Some sections and phrases also seem to be the same as pages 49-51 from this document https://id4ex.il.pw.edu.pl/wp-content/uploads/2022/06/ID4EX-IO1_v0.8.pdf

Author Response

Dear colleague, the cited article refers to the same work, but here the topics are discussed in a more in-depth and articulated way. It would be difficult to describe the same technologies in different linguistic ways. This article aims to be a more exhaustive and articulated treatment of the mentioned article in Italian. The ID4EX project page was only a communication of the technologies used, also in this case the project is the same, so unfortunately some explanations coincide.

In the meantime, I'm sending you an updated version with suggestions from Yolanda Pang.

FG

Round 2

Reviewer 3 Report

No further comments.